# Quantifying the rise and fall of scientific fields

Chakresh Kumar Singh[1,2], Emma Barme[1,2], Robert Ward[3], Liubov Tupikina[1,2,4], Marc Santolini[1,2]*

**1** Université Paris Cité, Inserm, System Engineering and Evolution Dynamics, Paris, France, **2** Learning Planet Institute, Paris, France, **3** School of Public Policy, Georgia Institute of Technology, Atlanta, GA, United States of America, **4** Nokia Bell Labs, Nozay, France

* marc.santolini@cri-paris.org

**Data Availability Statement:** The data underlying the results presented in the study are available from https://doi.org/10.5281/zenodo.6598737.

**Funding:** Thanks to the Bettencourt Schueller Foundation long term partnership, this work was partly supported by the CRI Research Fellowship to

## Abstract

Science advances by pushing the boundaries of the adjacent possible. While the global scientific enterprise grows at an exponential pace, at the mesoscopic level the exploration and exploitation of research ideas are reflected through the rise and fall of research fields. The empirical literature has largely studied such dynamics on a case-by-case basis, with a focus on explaining how and why communities of knowledge production evolve. Although fields rise and fall on different temporal and population scales, they are generally argued to pass through a common set of evolutionary stages. To understand the social processes that drive these stages beyond case studies, we need a way to quantify and compare different fields on the same terms. In this paper we develop techniques for identifying common patterns in the evolution of scientific fields and demonstrate their usefulness using 1.5 million preprints from the arXiv repository covering 175 research fields spanning Physics, Mathematics, Computer Science, Quantitative Biology and Quantitative Finance. We show that fields consistently follow a rise and fall pattern captured by a two parameters right-tailed Gumbel temporal distribution. We introduce a field-specific re-scaled time and explore the generic properties shared by articles and authors at the creation, adoption, peak, and decay evolutionary phases. We find that the early phase of a field is characterized by disruptive works mixing of cognitively distant fields written by small teams of interdisciplinary authors, while late phases exhibit the role of specialized, large teams building on the previous works in the field. This method provides foundations to quantitatively explore the generic patterns underlying the evolution of research fields in science, with general implications in innovation studies.

## Introduction

Quantifying the dynamics of scientific fields can help us understand the past and design the future of scientific knowledge production. Several studies have investigated the emergence and evolution of scientific fields, from the discovery of new concepts to their adaptation and modification by the scientific community [1–6], to the slow-down in their growth due to the ever rising number of publications [7]. In particular, methods ranging from bibliometric studies [8, 9] to network analyses [6, 10, 11] and natural language processing [12–14] have been

MS. Nokia bell labs provided support in the form of salaries for author LT, but did not have any additional role in the study design, data collection and analysis, decision to publish, or preparation of the manuscript. The specific roles of these authors are articulated in the 'author contributions' section.

**Competing interests:** The commercial affiliation does not alter our adherence to PLOS ONE policies on sharing data and materials.

implemented on large publication corpora to monitor the propagation of concepts across articles [15–17] and the social interactions between researchers that are producing them [11, 18–20].

To study field evolution, one first needs an operational definition of a field. Scholars have most often defined fields using taxonomies provided by professional societies (as in the case of PACS, the Physics and Astronomy Classification Scheme [21], the ACM Classification Scheme in Computer Science or the JEL scheme in Economics) that authors use to self-annotate their papers, or taxonomies used by citation indexers (as in the case of PubMed MeSH terms and Web of Science Subject Categories) to annotate papers in their collection. Taxonomies at the level of topics can then be clustered into fields. For example, when working with the more than 30,000 MeSH terms, scholars have defined fields by constructing their co-occurrence network and clustering dense areas of connection into discrete fields [10, 12]. Others have used citation networks themselves to group articles by relatedness and map the knowledge flow within and across research fields [6, 16, 22, 23]. Still others have inferred areas of research directly from the text of scientific papers, avoiding the manual annotations performed by authors and indexers elsewhere [13–15, 24]. These various methods provide clusters of closely related topics corresponding to putative research fields, allowing to monitor how the changing relations between topics and ideas underlie the dynamic evolution and mutual interactions between fields. A final stream of literature infers field structure from the collaboration networks of their members. For example, the co-authorship network between researchers has been shown to undergo a topological transition during the emergence of a new field [18, 19]. Co-authorship relations also influence the individual evolution of research interests and foster the emergence of a consensus in a research community [25–27].

While fields rise and fall on different temporal and population scales, they are generally argued to pass through a common set of evolutionary stages [1, 3], a process driven by consensus building within the scientific community and the discovery of new research fronts and the development of corresponding technologies [2, 28]. Different stages are typified by micro-level changes in membership and behavior proposed to drive macro-level dynamics from birth to death. To study these temporal patterns, dynamical models were introduced to characterize the evolution of research fields [8, 29] and the spread of innovation [30–32]. Yet, we are still lacking a unified empirical framework that delineates these stereotyped stages in the evolution of scientific fields and investigates the specific actors and behaviors across them that can be validated over a large number of well-annotated research fields.

Here, we address this gap by developing techniques for identifying common patterns in the evolution of fields. We demonstrate their usefulness using a large corpus of 1.45 million articles from the arXiv repository with self-reported field tags spanning 175 research fields in Physics, Computer Science, Mathematics, Finance, and Biology. We show that the evolution of fields follows a right-tailed distribution with two parameters characterizing peak location and distribution width. This allows us to collapse the temporal distributions onto a single rise-and-fall curve and delineate different evolutionary stages of the fields: creation, adoption, peak, early decay, and late decay. We then describe the characteristics of articles and authors across these stages. We finish by discussing these results and their implications for further work in science and innovation.

## Results

### Description of the data

Since its launch in 1991, the arXiv repository has become a major venue for research dissemination of particular importance in fields of Physics, Mathematics and Computer Science. As

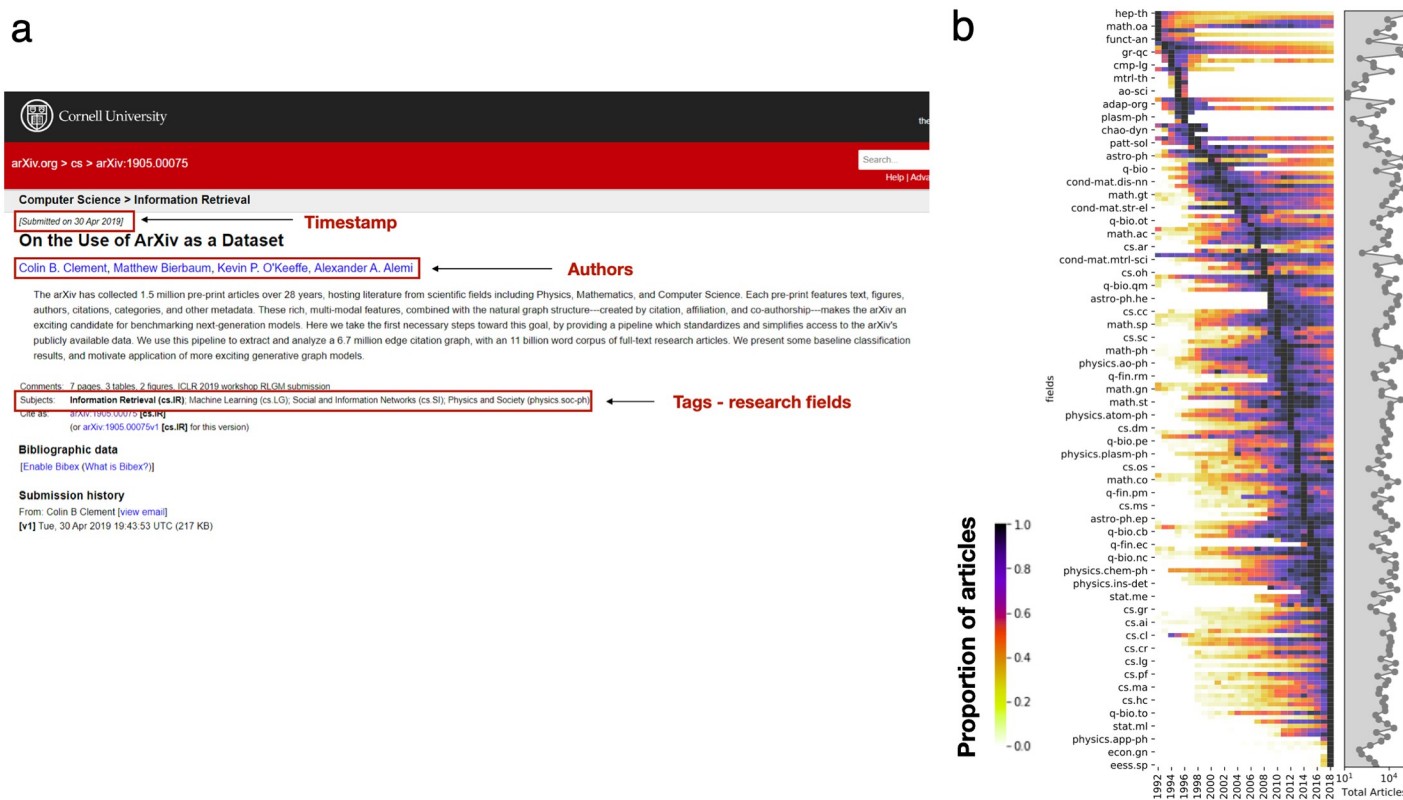

**Fig 1. arXiv dataset. a** Example of an article in arXiv, highlighting the metadata extracted using the arXiv API. **b** Heatmap representing the share of articles in each field (rows) over time (columns). Field are identified using the subject tags within articles. The heatmap is row normalized for comparison across fields. Fields are ordered in chronological order of their peak time. The right side panel shows the total number of articles published in each field in log scale.

an open and free contribution platform, it provides an equal opportunity for publication to researchers globally, and plays a dominant role in the diffusion of knowledge [33] and the evolution of new ideas [16].

When submitting a contribution, authors declare the research fields that the article is contributing to by selecting from a list of subject tags. Here we collected information about authors, date of publication, and research fields of 1,456,403 arXiv articles until 2018 (see Methods section and Fig 1a). The number of articles and authors exhibit an exponential growth over time with a doubling period of 6 years (see Fig 1b and S1 Fig in S1 File). To control for this effect, here we focus for each field $i$ on the yearly share of articles $f_{i,y} = n_{i,y}/N_y$, where $n_{i,y}$ is the number of articles published in the considered field at year $y$ and $N_y$ is the total number of articles in arXiv in the same year. We represent the temporal distributions of all fields in Fig 1c by chronological peak time. Over the past 30 years, the research interests have shifted from high-energy physics to computer science and more recently economy.

## Quantifying the rise and fall of scientific fields

Despite differences in overall number of articles and eventual duration, we observe a general rise-and-fall pattern across research fields (Fig 1c), prompting us to explore if a simple model can capture their temporal variation.

Extreme value theory [34] predicts that under a broad range of circumstances, temporal processes displaying periods of incubation (such as incubation of ideas) or processes with

multiple choice (such as the choice of ideas or research fields) follow skewed right-tailed extreme value distributions. Examples of such processes can be found in diverse areas, for example when modelling the evolution of scientific citations [35] or disease incubation periods [36, 37]. In the context of the evolution of scientific fields, epidemiological models have been proposed [29] to capture the dynamical patterns underlying the contagion of ideas and the choice of researchers to work on different ideas. Such models exhibit right-skewed temporal distributions in the number of infected individuals, which in this case consists of articles and researchers in the field.

Following these insights, here we use the Gumbel distribution (Eq 1) as an ansatz to model the observed field temporal distributions. Belonging to the general class of extreme value distributions [34, 38], it provides interpretable parameters for the peak location $\alpha$ and distribution width $\beta$ (S3 Fig in S1 File). Denoting by $t$ the time since the first article was published in the field, the share of articles $G(t)$ follows Eq 1:

$$G(t) = \frac{1}{\beta} e^{\frac{-(t-\alpha)}{\beta}} e^{-e^{\frac{-(t-\alpha)}{\beta}}} \tag{1}$$

where $\alpha$ is the location parameter and $\beta$ the scale parameter.

In order to estimate the model fit, we consider fields that satisfy three conditions as represented in Fig 2a: (i) longevity—having at least 10 years of activity to ensure a sufficient observation period, (ii) unimodality—we exclude multimodal distributions as it would require introducing a mixture model going beyond the scope of this study and (iii) completeness—we require the peak of the distribution to be at least 3 years away from the beginning and the end of the collection period to ensure that we capture sufficient data on both sides of the distribution. This reduces the number of fields to 72, which we consider in our analyses below.

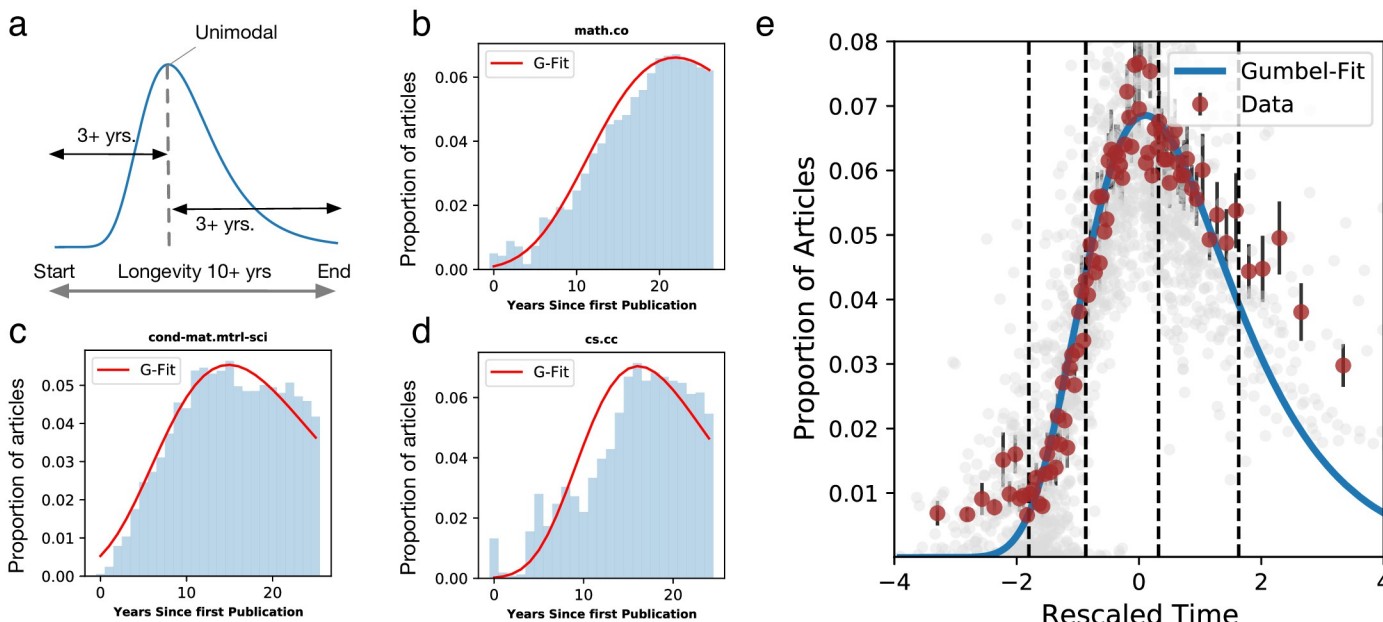

**Fig 2. Temporal evolution of fields. a**. Conditions for a field to be included in the analysis. **b-d** Gumbel fits for the fields with the largest numbers of articles in Physics, Mathematics and Computer Science: Mathematics—Combinatorics (b), Material Science (c) and Computational Complexity (d). **e**. Evolution of the 72 studied fields after temporal re-scaling from Eq 2. The blue curve represents the Gumbel fit, and red dots correspond to the empirical average over equal sample-size bins. Error bars indicate standard error.

Using a least-squares optimization fitting procedure (see Methods), we show that 66 out of 72 fields (91.6%) exhibit a significant goodness of fit ($k < 0.3$ and $p > 0.05$ under KS-test). We show in Fig 2b–2d the temporal distributions and Gumbel fits for the fields with the largest total numbers of articles in Physics, Mathematics and Computer Science. For every field we fit the corresponding $\alpha$ and $\beta$ parameters. To get more insights on the interpretation of the $\alpha$ and $\beta$ parameters, we show in S3a, S3b Fig in S1 File some examples of Gumbel distributions with varying parameters. The location parameter $\alpha$ corresponds to the mode of the distribution, i.e when fields peak in their lifetime, while the scale parameter $\beta$ corresponds to the distribution width. Fields with a low $\beta$ have a rapid rise followed by a rapid decay with a long tail. These could be the fields promoted by sudden advances in science and technologies or economics, for example, the field of Pricing of Securities in Quantitative finance (q-fin.pr) (S3c Fig in S1 File). On the other hand fields with a large $\beta$ have an elongated rise and fall with a long tail in the decay phase—for example Condensed Matter Material Sciences (cond-mat.mtrl-sci) (S3d Fig in S1 File).

After obtaining the location $\alpha$ and scale $\beta$ parameters from the fitting procedure, we compute for each field the re-scaled time:

$$t' = \frac{t - \alpha}{\beta} \tag{2}$$

By re-normalizing fields with this standardized time, we observe that the various temporal distributions align on a single curve, highlighting the shared patterns of rise and fall across the fields studied (Fig 2e). In particular, the Gumbel distribution provides a more stringent fit of the tails, as can be observed when comparing to other distributions (S4 Fig in S1 File).

## Characterizing the stages of research field evolution

Using the re-scaled time from Eq 2, we next explore the characteristics of articles and researchers at different stages of a research field evolution. We adopt hereafter the standard delineations of stages from the innovation diffusion literature [30] and define 5 periods of research field evolution (creation, adoption, peak, early decay, and late decay) delineated on the re-scaled timeline corresponding respectively to the 2.5%, 16%, 50% and 84% quantiles of the Gumbel distribution in Fig 2e (blue curve). We then group articles within these categories for each field and examine the variation of their characteristics when averaging across all fields.

We consider characteristics of the articles submitted at various field stages, and of the authors who submit them. For articles, we focus on the number of fields reported (article multidisciplinarity), the number of authors (team size), the number of references made to other arXiv articles, and the number of citations received within arXiv (article impact). For authors, we consider their career stage within arXiv at the time of submitting the article, the total number of articles submitted to arXiv (longevity), the number of fields their articles span during their career and the number of fields per article (author multidisciplinarity). We average these characteristics over the article coauthors for which we have a unique identifier (ORCID). In the case of career stage $s$, we use Eq 3, where $N_{art}$ is the chronological rank of the current article across the author's publications and $N_{tot}$ is the total number of articles:

$$s = \frac{N_{art} - 1}{N_{tot} - 1} \tag{3}$$

We show in Fig 3 the average values of these features for each stage across the 72 fields along with random expectations (see Methods). In the context of article metrics (Fig 3a), we find that the early stages of research fields are characterized by interdisciplinary articles (2.36 fields, vs. 2.05 for late decay) co-authored by small teams (2 authors vs 4.5). As fields evolve,

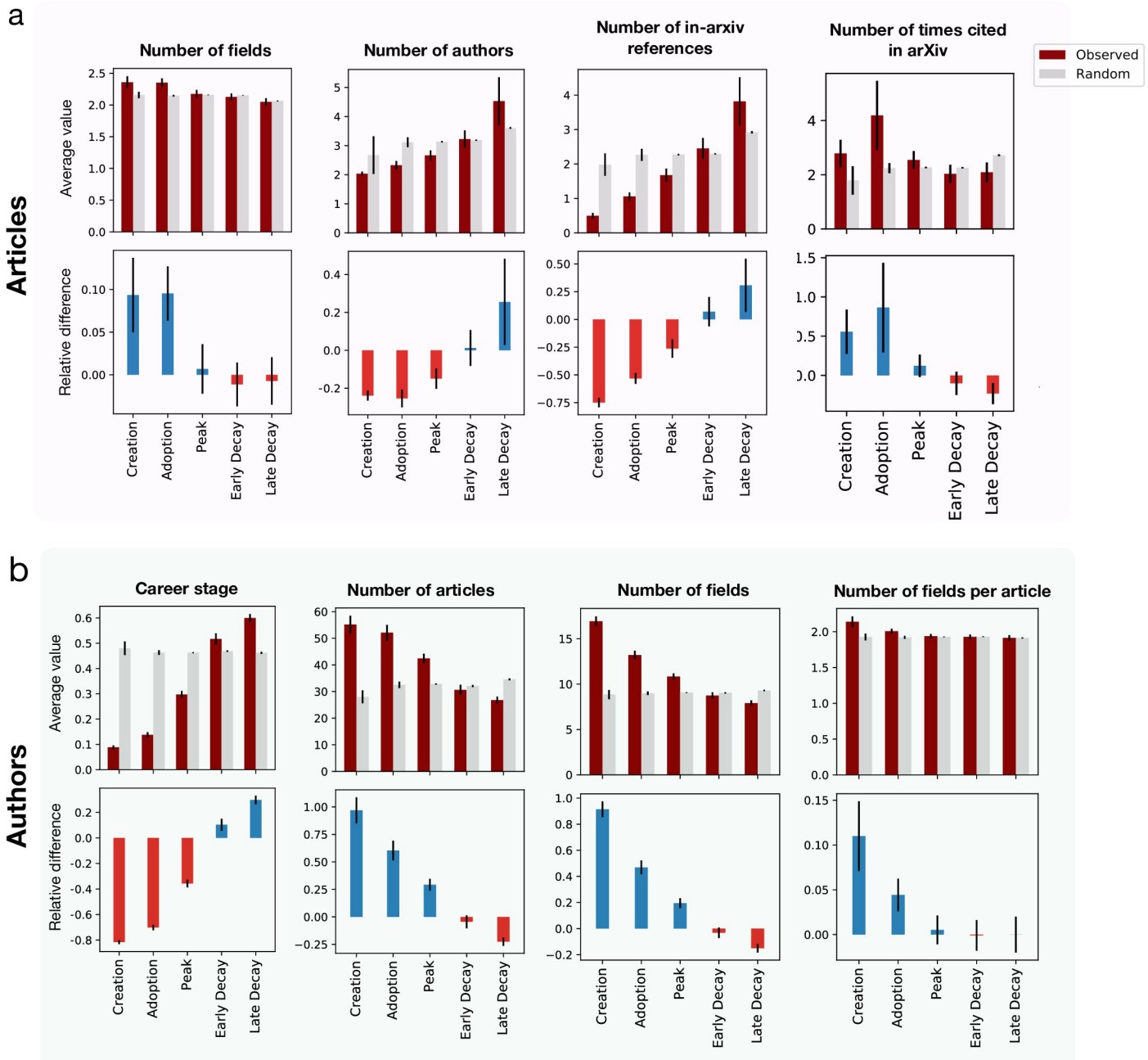

**Fig 3. Evolution stages.** Characteristics of articles and authors at different evolutionary stages. The observed values are averaged over all fields (red bars). Gray bars correspond to the average field-specific random expectation (see Methods). Bottom plots represent the relative difference between observed and random values. Error bars denote standard errors for observed values (red) and standard deviation for random values (gray). **a** Article-centric features: number of fields reported in the article (multidisciplinarity), number of authors (team size), number of references made to other arXiv articles, and number of citations received within arXiv (impact). **b** Author centric features: career stage within arXiv at the time of submitting the article, total number of articles submitted to arXiv (longevity), number of fields their articles span during their career and average number of fields per article (multidisciplinarity).

we observe a steady growth in the number of references to earlier arXiv articles, indicating that the community builds on earlier works in arXiv (Fig 3 and S5a Fig in S1 File when restricting to the same field). Finally, we find that article impact, measured by the number of citations within arXiv, is maximal at the Adoption phase before the field has reached its peak. The citation count observes a similar trend in the case of total citations within arXiv shown in Fig 3a as well as citations within arXiv received in the first five years (S5b Fig in S1 File). For author metrics (Fig 3b), we find that the early stages of research fields are characterized by multidisciplinary authors (16.9 fields in career for creation vs 7.9 in career for late decay, and 2.13 fields per article vs 1.91 fields per article), who tend to be in their early career (8% of total duration vs 60%) with the longest longevity (55 papers vs 27).

These observations show that small teams of interdisciplinary authors contribute to the early stages of a field's evolution, in line with previous findings that small teams produce more disruptive [39], novel [40] and creative research [41]. To test these insights further, we calculated the disruptive index of all arXiv papers using the internal citation network (see Methods). We find that papers in the creation stage have the highest disruptive index, followed by a steady decrease in later stages (Fig 4), highlighting the relevance of the delineated phases as markers of innovation and development of a field.

### Cognitive distance and early innovation

The previous results show that works submitted in early phases of research fields tend to mix a larger number of field tags. However, this measure does not take into consideration the various levels of similarity between fields. For example, publishing an article within sub-fields of physics is different from publishing an article mixing quantitative biology, computer science, and physics. This is rendered apparent when examining the co-occurrence network of fields across arXiv articles (Fig 5a). In the co-occurrence network, nodes represent field tags, and edges represent their co-occurrence across articles. The network represents the landscape of fields in the arXiv, with closely related fields clustering together into communities corresponding to 6 broader categories: Physics (purple), Quantitative biology (gray), Computer Science (green), Mathematics (blue), Statistics (pink) and Quantitative Finance (orange).

Using this network construction (details in Methods), we define the Cognitive distance $C_{i,j}$ between field tags $i$ and $j$ as the weighted shortest path $C_{i,j} = \sum_e \frac{1}{W_e}$, where $e$ are the edges on the shortest path between the two tags $i$ and $j$ and $W_e$ are their weights in the co-occurrence network. This cognitive distance allows us to provide a weighted proxy for interdisciplinarity. In particular, it allows to quantify the distance between disconnected fields: an example of this is shown in S6 Fig in S1 File where q-fin.ec (Economics) connects to hep-ph (High Energy Physics) by a path length of 4.

We use this measure to compute for each article with at least two field tags the maximum cognitive distance between any pair of tags. We find that articles published in the early stages of a research field have a significantly larger cognitive distance, while the measure decays to the random level by peak stage (Fig 5b). Similarly, for authors we find that in earlier stages authors publish in cognitively distant fields, which narrows down to similar fields in later stages (Fig 5c). The relative difference with random at the creation stage is more stringent than the previous measure using number of tags (articles: 0.3 vs 0.1, authors: 0.8 vs 0.1), strengthening our previous observation.

### Discussion

Science evolves as scientists cast and redirect their attention over the knowledge space. As scientists are drawn to similar topics, collective attention organizes into fields of inquiry.

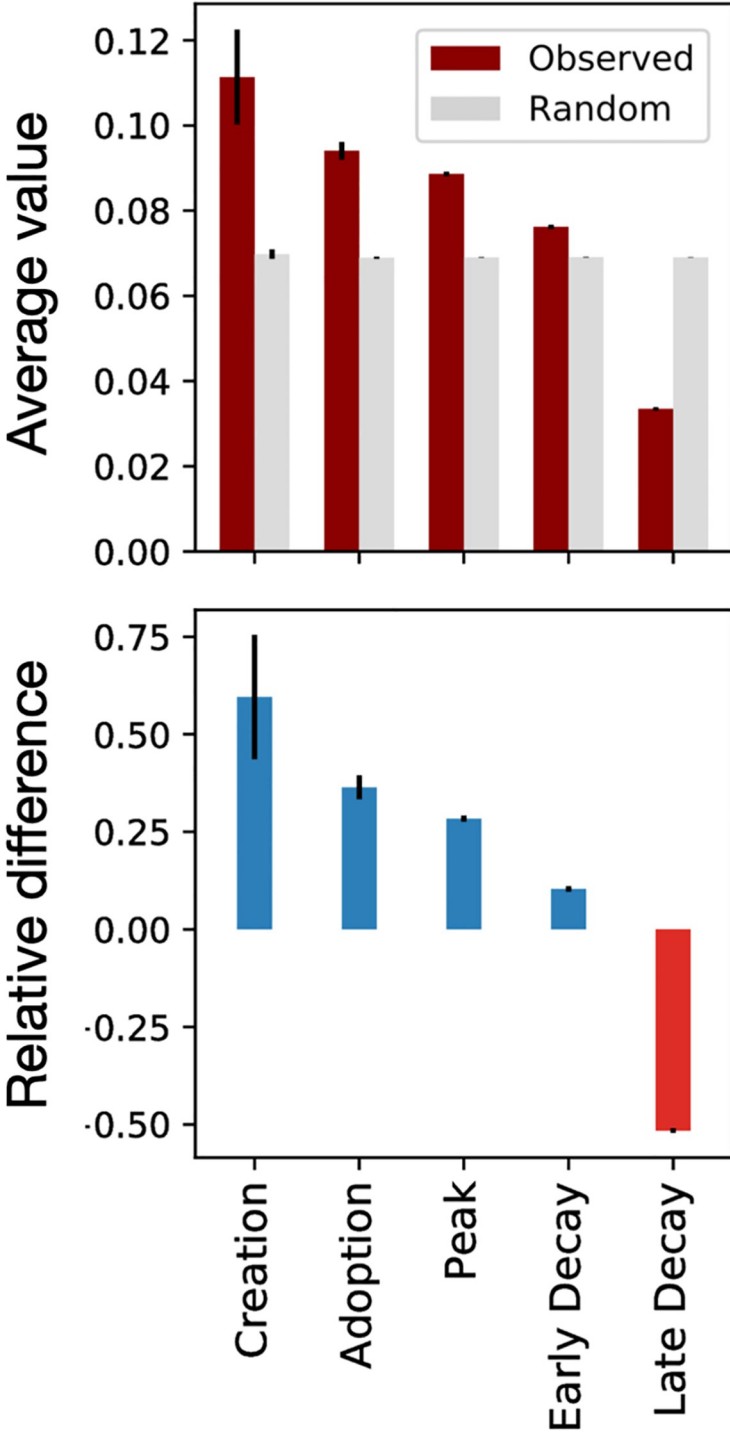

**Fig 4. Disruptive index.** Disruptive index measure for the articles in arXiv. The observed values are averaged over all articles (red bars). Gray bars correspond to the average random expectation (see Methods). Bottom plots represent the relative difference between observed and random values. Error bars denote standard errors for observed values (red) and standard deviation for random values (gray). Earlier stages of fields evolution have more disruptive articles with a steady decrease in later stages.

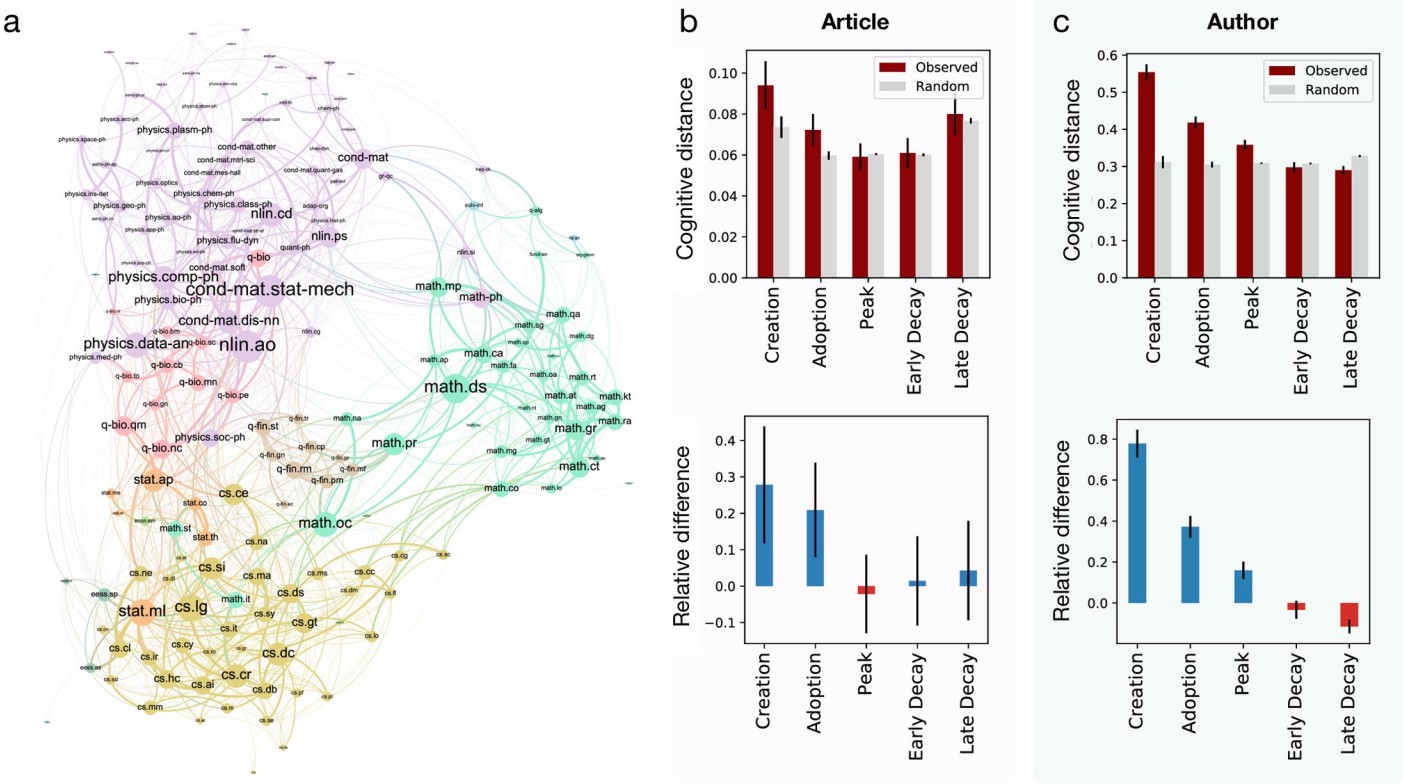

**Fig 5. Cognitive distance. a** Co-occurrence network of arXiv field tags. Nodes are colored based on the major research area they belong to (Physics, Computer Science, Mathematics, Statistics, Quantitative Finance, Quantitative Biology). Barplots in **b,c** follow the same method than in Fig 3. **b** Average cognitive distance across the field tags of articles. **c** Average cognitive distance across all the field tags used by authors throughout their career.

Understanding how fields emerge, grow to maturity and decline is thus a central challenge for the science of science [42] but the backing for claims about how and why fields evolve has historically consisted illustrative cases rather than comprehensive evidence. Although theoretical claims are often universal—applying equally to large and small, fast and slow changing fields—our ability to make comparisons across different temporal and population scales has been limited, making it difficult in turn to provide rigorous tests of such claims.

In this study we introduced a method for rescaling the dynamics of scientific fields to normalized population and temporal scale. We argued that the processes that drive field evolution are likely within the family of extreme value distributions and showed that 91% of the fields in our sample are well fitted by a Gumbel distribution. To demonstrate the utility of this method, we applied it to 1.5M articles from the arXiv preprint repository and compared the membership and behavior of authorship teams at different canonical stages of field evolution. Our descriptive results suggest common patterns consistent with existing theory. Early evolutionary stages are populated by small teams of early career researchers from diverse and dissimilar disciplines. As the number of papers in a field saturates, we observe a shift to larger and more specialized teams. These organizational shifts are coupled to shifts in the type of knowledge being produced. Early stages abound in disruptive articles—contributions that replace prior knowledge in the citation graph—with that disruptiveness fading as a field evolves. We find that the highest impact papers too are produced before the field reaches maturity, perhaps reflecting the publication of foundational texts as a field is born that most subsequent authors

cite. Finally, we observe authors becoming less reliant on references to external fields as a new field matures, in line with the argument that fields internally integrate their knowledge over time [43].

Our study is not without limitations. We focus on fields whose dynamics are unimodal—with a single rise and fall—which limits our sample to 72 out of the 175 present on arXiv. Follow on work can extend the method introduced here by considering mixture models capable of capturing multiple waves of interest in a field. When estimating the age composition of author teams, we relied on ORCID IDs to disambiguate author names, which similarly limited our sample to a small and biased subset of authors. For the Disruptive Index measure we note that the citation network in our dataset only includes citations within arXiv, which are sparser compared to Web of Science or Scopus for example, and citation behavior may be particular to the fields predominant on arXiv or variant across them. While arXiv is an open-access repository, submitting authors need to be invited by another existing member from the main field of interest. As with other venues, gatekeeping and the ability of authors to effectively navigate it shapes who can publish and therefore membership in the fields of our sample [44, 45]. Finally, the descriptive results shown in this study should not be interpreted as causal nor should they be taken as representative of science as a whole. Future work should expand this study to larger, more representative databases with more comprehensive paper and author level metadata.

More comprehensive fields-level data could also help understand broader trends within the arXiv community as a whole. For example, while originally focused on Physics fields, arXiv submissions have become more diverse, with increasing submissions from the Computer Science and Economics fields (Fig 1b). This uptake could be related to an increase in research demand from the side of these fields (problems, grants, and publication opportunities), or research supply from the side of Physics (e.g., talent spillover from physics, which attracted and trained capable researchers but does not necessarily offer enough relevant jobs). Alternatively, this can be part of a broader trend towards open access research and increasing leniency towards preprints on the part of publishers [46]. Institutional factors at the level of funding agencies or universities, as well as norms within particular organizations or fields may also create field-specific incentives to share open access versions of papers. Future work could study what drives this trend, for example by examining formal requirements for open access and using surveys of scientists in these fields and others to understand the preferences and incentives that drive this behavior.

Beyond these considerations, we hope the methods introduced here help future studies build a deeper understanding of scientific change. First, the composition and behavior of scientific fields at different stages of their evolution bears deeper examination, in particular with the goal of understanding the causes of birth, maturity and decline and the extent to which these drivers are universal or particular. Many scholars have argued that these causes are scale-invariant. For example, in his postscript to The Structure of Scientific Revolutions, Kuhn clarifies that the emergence of a new paradigm "Need not be a large change nor need it seem revolutionary to those outside a single community, consisting perhaps of fewer than twenty-five people." [1]. At the same time, scaling effects involving nonlinear relationships between size and collective behavior are common in complex systems. The effects of scale on scientific change remain unresolved.

Broad evidence suggests the social organization of science is changing, yet how these changes affect field evolution is unclear. In particular, the trend towards increasing team size [47, 48] and bureaucratization [40, 49] bears consideration. Does the size and structure of a team affect their propensity to enter a field at each stage? Or their capacity to succeed, conditional on entry? Are the drivers of field evolution stationary? Or have they changed with the

micro-organization of science? Additionally, while there is growing work on the effect of funding size and structure on individual scientists, [50–52] the optimal ways to grow a new field remain unclear.

Finally, while scientific knowledge evolves as a function of scientists decisions, it also feeds back into the decision making process. That is, the structure of scientific knowledge shapes which problems are salient and solvable, constraining scientists future behavior [1, 53]. These effects seem likely to result from both first-order structure (the patterns in which individual concepts are connected) as well as higher-order structure (the patterns in which sets of concepts are connected). Some neighborhoods of the knowledge space are presumably more fertile ground for new fields to grow than others. It may also be true that the neighborhood in which a field lies is a key condition that shapes its dynamics far into the future. Recent work has found that the sheer volume of new papers published in a field is a strong predictor of disruption and growth [7]. Similarly, scholars have begun to study how the higher-order structure of scientific knowledge evolves in time [54]. Future work can probe deeper into exactly how the structure of scientific knowledge shapes field evolution.

Overall, explaining and predicting the evolution of scientific fields may require models including all of the factors discussed—sensitivity to scale, to time, to the organization of scientists and of scientific knowledge—along with the interactions between them. By making cross-field comparisons easier to make, our method should prove as useful for studying the role of single factors as for testing more complex models of field evolution. As such, we expect these insights to be helpful for researchers and policymakers interested in the emergence and development of research fields and more broadly in the dynamics of innovation.

## Methods

### Dataset extraction

We extracted the publication metadata from the arXiv website using the arXiv API (https://arxiv.org/help/api/). The data spans years 1986 to 2018, with a total of 1,456,404 articles. For each article we retrieved the following characteristics: a) the unique article ID, b) the timestamp of article submission, c) the list of subjects categories (field tags), d) the citations received within arXiv, e) the references to other arXiv articles, and f) the list of last names of authors. We show an example article in Fig 1a. Furthermore, we extracted when possible the ORCID IDs of the authors that declared it in arXiv. The number of unique ORCID IDs was 50,402, allowing to disambiguate these authors' names. The dataset produced can be accessed on https://doi.org/10.5281/zenodo.6598737. For the citation network we used the dataset from [55] which can be accessed on https://www.kaggle.com/datasets/Cornell-University/arxiv or https://github.com/mattbierbaum/arxiv-public-datasets/releases.

### Fitting procedure

**Uni-modality test.** For filtering multi-modal fields we use the *diptest* R library to compute the dip unimodality test. We remove fields that fail the test ($p < 0.05$). Separately, we also use the kernel density estimation(*kde*) method to find the peaks using varied bandwidths. Fields that have consistent bimodality are removed. Finally, fields that are unimodal in both the calculations are chosen.

**Least square optimization.** For the selected fields, we strip years before the first publication to only consider years since first article. We then constrain the mode of the fitted distribution to coincide with the empirical one, and we fit the location and scale parameters using least-square optimization.

## Assigning articles and authors to evolutionary stages

We first collect for each field all articles containing the field tag. We associate each article to the evolutionary stage corresponding to the re-scaled time obtained for that particular field. We then assign the authors of each article with an ORCID ID to the corresponding evolutionary stage. Note that articles with multiple field tags can be assigned to different stages of evolution corresponding to the re-scaled times of the different tags.

## Randomization

The observed features in Fig 3 are compared with random expectation by shuffling for each field the re-scaled times across articles. This procedure is repeated 50 times for each field and we compute the average for each stage. Finally, we compute the average and standard deviation across fields.

## Disruptive index measure

Given a paper $p$ followed by $N_s$ subsequent works, if $N_i$ papers out of the $N_s$ cite only $p$, $N_j$ out of $N_s$ cite both $p$ and its references and $N_k$ out of $N_s$ cite only its references, then the Disruptive Index (DI) of the paper $p$ [39, 56] is defined by

$$DI_p = \frac{N_i - N_j}{N_s} \tag{4}$$

where $N_s = N_i + N_j + N_k$.

## Constructing tags co-occurrence network

We construct an undirected, weighted tag co-occurrence network where the edge weights indicates the similarity between the fields corresponding the tags. Let $N$ are the total number of articles published, $K$ the number of articles using field tag $i$ and $n$ the number of articles using field tag $j$ with $k$ articles that use both $i$, $j$. Then integrating Eq 5

$$p_{ij} = \frac{\binom{K}{k}\binom{N-K}{n-k}}{\binom{N}{n}} \tag{5}$$

for $k$ or more articles yields the hypergeometric p-value $p_v$ that the two fields have at least this number of co-occurrences given the number of times they each have occurred. Note that here lower p-values indicate stronger similarity. As such, we define the edge weight $W_{ij}$ between fields $i$ and $j$ as $-log_{10}(p_v)$. Edges corresponding to $p_v > 0.01$ are filtered out.

## Supporting information

**S1 File.**
(PDF)

## Author Contributions

**Conceptualization:** Liubov Tupikina, Marc Santolini.

**Data curation:** Chakresh Kumar Singh, Emma Barme.

**Formal analysis:** Chakresh Kumar Singh, Marc Santolini.

**Funding acquisition:** Marc Santolini.

**Investigation:** Marc Santolini.

**Methodology:** Chakresh Kumar Singh, Liubov Tupikina, Marc Santolini.

**Project administration:** Marc Santolini.

**Supervision:** Marc Santolini.

**Validation:** Chakresh Kumar Singh, Marc Santolini.

**Visualization:** Chakresh Kumar Singh, Marc Santolini.

**Writing – original draft:** Chakresh Kumar Singh, Robert Ward, Liubov Tupikina, Marc Santolini.

**Writing – review & editing:** Chakresh Kumar Singh, Robert Ward, Liubov Tupikina, Marc Santolini.

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
