## [Decision Letter · Decision Letter 0]

4 Jan 2022

PONE-D-21-36265Quantifying the rise and fall of scientific fieldsPLOS ONE

Dear Dr. Santolini,

Thank you for submitting your manuscript to PLOS ONE. After careful consideration, we feel that it has merit but does not fully meet PLOS ONE’s publication criteria as it currently stands. Therefore, we invite you to submit a revised version of the manuscript that addresses the points raised during the review process.

The two Reviewers agree that this manuscript is worth of publication after some issues are addressed - I agree with their assessment and encourage the authors to revise the manuscript according to the Reviewers' points and resubmit. 

We look forward to receiving your revised manuscript.

Kind regards,

Roberta Sinatra

Academic Editor

PLOS ONE

Journal Requirements:

“The authors have declared that no competing interests exist.”

We note that one or more of the authors are employed by a commercial company: Nokia Bell Labs.

“Thanks to the Bettencourt Schueller Foundation long term partnership, this work was 240 partly supported by the CRI Research Fellowship to Marc Santolini.”

Reviewers' comments:

Reviewer's Responses to Questions

**Comments to the Author**

1. Is the manuscript technically sound, and do the data support the conclusions?

Reviewer #1: Yes

Reviewer #2: Yes

2. Has the statistical analysis been performed appropriately and rigorously? 

Reviewer #1: Yes

Reviewer #2: Yes

3. Have the authors made all data underlying the findings in their manuscript fully available?

Reviewer #1: Yes

Reviewer #2: Yes

4. Is the manuscript presented in an intelligible fashion and written in standard English?

Reviewer #1: Yes

Reviewer #2: Yes

5. Review Comments to the Author

Reviewer #1: Summary

This paper reveals the common pattern in research productivity over the lifecycle of academic fields by analyzing millions of papers submitted to arXiv, the open-access repository of digital preprints, over two decades (Fig. 1). Annual research productivity in a field changes over time, following a growth curve of “fast increase and slow decay (Fig. 2),” during which process small teams dominate the first stage by combining distant, interdisciplinary ideas to create theories, methods, and paradigms, pushing the field towards the peak of productivity, whereas large teams dominate the second stage by citing, developing, and fine-tuning what has been created, leading to the maturity and eventually decline of the field (Fig. 3).

Contributions

The contribution is threefold 1) identifying the common patterns of productivity lifecycle across fields, which allows for the effective comparison between fields of different history length and distinct productivity levels in search of universal mechanisms of attention shifting and knowledge production; 2) linking team characteristics and contribution to field development, demonstrating how small and young teams working on broad topics are followed by large and old teams working on specialized topics; 3) quantifying the cognitive distance between fields to support team interdisciplinary analysis, which eases the concerns that some fields are closer than others.

Recommendation

The method is solid, the writing is fluent, and the organization of main figures and supportive materials is clear. Meanwhile, the introduction and discussion sessions can be strengthened. I recommend major revision. My assessments and suggestions for which I expect point-by-point responses are shown below.

Assessments and suggestions

I used the following checking items to guide my assessment in addition to the journal’s rubric.

1. Research question non-trivial?

Yes. The pattern of field development and its association with individual research choices is an important question with both theoretical meanings (organization of scientific knowledge) and policy implications (national funding strategies to develop fields).

2. Assumption rationale insightful?

No. A convincing assumption rationale is not presented. The current rationale reads like “because Gumbel is used a lot elsewhere so we assume it may also work here” (line 73-74). I would encourage unfolded discussions on why “fast increase and slow decay” should be anticipated if Gumbel is pre-determined as the best candidate. Otherwise, Gumbel is discovered rather than expected, then alternative distributions should be compared. The best is to do both.

3. Assumption testing solid?

Yes, but could be better. The authors only calculate the goodness of fit for the Gumbel distribution rather than comparing Gumbel against other possible candidates. For example, why is Poisson distribution, which is similarly skewed but has fewer parameters, not considered? Without comparing against other distribution of similar shapes, it is hard to understand what the Gumbel model uniquely contributes. While the distribution fitting and comparison can be improved, the comparison against random-shuffle models in presenting the association between team characteristics and field stage is well done (Figure 3).

4. Literature review relevant and extensive?

Yes, but weak. The author claimed “we are still lacking a unified framework to delineate stereotyped stages in the evolution of scientific fields that can be validated over a large number of well-annotated research fields. (Line 33)” This is not true. While I agree that large-scale empirical tests are rare, theoretical framework on the development of scientific fields is not rare. Randall Collins’s theory on “high-consensus, rapid-discovery science” (1994, https://link.springer.com/article/10.1007/BF01476360 ) is the most relevant to this study. Also the work by Moody & White (2003, https://www.jstor.org/stable/3088904 ) in analyzing field cohesion. Recent work measuring the “paradigmaticness” of disciplines by Evans et al. (2016, https://sociologicalscience.com/articles-v3-32-757/ ) from an NLP perspective could also be a good complement to the network perspective of this work.

5. Large and novel datasets that may contribute to the community?

Yes, but could be better. The author used arXiv which is open-access datasets and specified the API in the Methods section (URL should be provided). The best is to also share processed datasets to maximize the impact of this work through supporting future explorations.

Additional suggestions:

6. If the authors want to maximize audience appreciation, maybe want to consider getting rid of “scale invariant?” This term comes from physics where pattern universality across scales is appreciated as a default. For scholars from bibliometrics, information science, policy scholars, and more, who do not necessarily have this background but could potentially be interested in this paper, this word may push them away. The use of this word implies that (field) size is the main “trouble” in understanding knowledge organization so that if the size is controlled, the major challenge is solved. But large and small fields may have distinct dynamics that not necessary can or should be cheaply “controlled”. See recent discussion on slowed canonical progress in large fields by Chu & Evans (2018, https://www.pnas.org/content/118/41/e2021636118 ). I would encourage considering sticking to “common patterns,” but this is a rather personal (writing-style) suggestion that authors should feel free to ignore.

7. Somewhere in the paper, the Diffusion of Innovation Theory by Rogers (1962) (https://en.wikipedia.org/wiki/Diffusion_of_innovations) should be cited. Rogers is well-acknowledged by proposing the “s-curve” theory of diffusion and firstly identified innovators, early adopters, early majority, late majority, and laggards, which seem very similar to the five stages proposed by the authors.

8. The shift peak of arXiv research interests from physics to computer science and economy is worth saying more (line 64). Is it because of the increase in research demand (problems, grants, and publication opportunities) or caused by the increase in research supply (e.g., talent spillover from physics, which attracted and trained capable researchers but does not necessarily offer enough relevant jobs)?

9. The meaning of parameters alpha and beta should be discussed in the field development context (line 78).

10. The method authors used to model cognitive distance between fields is a network model (within which what the edge weight represented should be explained) but not “network embedding (line 153),” which is to learn the vector presentation of nodes and edges by extending word2vec (Mikolov, 2013). This differentiation is important because there is emerging work on quantifying cognitive distance between fields using network embeddings, such as (Peng et al. 2021: https://www.science.org/doi/10.1126/sciadv.abb9004) and (Lin et al., https://arxiv.org/abs/2103.03398).

11. The thresholds used to identify five stages seems arbitrary: “2.5%, 16%, 50% and 84% (line 104).” How these numbers are picked against “natural options” like 20%, 40%, 60%…should be justified. These numbers are very similar to the five numbers used by Rogers (https://en.wikipedia.org/wiki/Diffusion_of_innovations). If so, Rogers’ book should be cited with numbers explained.

12. How the author’s career stage is associated with the field stage should be explained better. For papers, it is straightforward to understand which field stage it belongs to because a paper has a publication year. But many scholars have a career length equal to or even longer than the analyzed time period (two decades).

Reviewer #2: This paper tackles the question on quantifying the dynamics of various scientific fields. The question is addressed by performing a statistical analysis and modeling of a dataset of well-annotated fields tags in Computer science, Mathematics, Finance and Biology, where for 1.45 million articles from 175 research fields in arXiv repository are used. The authors use Gumbel distribution as a proxy to model the field temporal distributions. It is a new and simple scale-invariant methods in quantifying evolution of scientific fields. Then the statistical analysis shows the general characteristics of articles and authors across evolution stages of fields: the early phase is represented by the mixing of cognitively distant fields by small teams of interdisciplinary authors, while late phases exhibit the role of specialized, large teams building on the previous works in the field.

The method and results are presented well using appropriate metrics and figures. The claims of the paper are supported by the results.

Everything looks technically correct and the language is clear.

I have two major issues and a few minor concerns regarding the analysis and interpretation of some of the results:

Major issues:

1. In order to control the growth effect over time of arXiv articles, the authors use yearly proportion of articles of considered field in arXiv as a metric to delineate the temporal characters instead of absolute number. My concern on this metric is that papers in some big fields will be like to be overwhelmed to small fields resulting in the misunderstanding of real evolution trend of small fields. For example, we can see from fig 2d, Computer science recently enters in decay stage. However, the empirical intuition is that computer science is booming specially in AI field. The phenomenon could also happen in other fields. The authors are suggested to further clarify the reasonable usage of this metric to eliminate reader’s worries.

2. By statistical analysis highlighting characteristics shared by articles and authors across various stages of a field evolution, the authors draw one of the main conclusions: early stages are characterized by small teams and interdisciplinary authors, while late stages exhibit the role of large, more specialized teams. They wrote in Discussion section of the paper “This supports the general finding that small teams disrupt while large teams develop science and technology [33, 34].” Although I agree the narration above, I also suggest that the authors make an extra effort to calculate the Disruptiveness Index to support their findings.

And there is a recently published paper closely related to the topic in this paper, it is recommended to be added in the reference list.

Johan S. G. Chu, James A. Evans. (2021). Slowed canonical progress in large fields of science. Proc Natl Acad Sci, 41(118), e2021636118.

A few minor concerns:

3.The paper uses re-scaled Gumbel distribution to model the rise and fall of a scientific filed. I have a detailed doubt about the calculation of stages. In fig 2b and fig 2d, if they use the 25%, 16%, 50% and 84% quantiles of the Gumbel distribution as cutpoints of stage, the math is recently on the decay stage. But at 84% quantiles, it is around the peak rather than late decline in fig 2b. Maybe I miss the details about the method. The authors are suggested to explain this point clearly.

4.Under strict conditions for data: longevity, unimodality and completeness, 175 research fields are reduced the number of fields to 72. The exiting rate of fields is bigger than ½, which means the authors lose and neglect lots of information in the main text. In my opinion, while it is much meaningful to find the scale-invariant patterns in the evolution of many scientific fields, the manuscript can be better if the authors clearly highlight the limitation in Discussion. The evolution characters of 72 fields in the paper cannot stand for the pattern of all fields in science. And it is worth exploring the untrivial patterns in further research.

5. In fig 1b, the authors compare the publishing seasonality between weekdays and weekends while the information is not mentioned or discussed at all to support the results in the manuscript. In my opinion, it is redundant and useless for publication.

6. The Fig 2d is lack of description in caption of Fig 2.

6. PLOS authors have the option to publish the peer review history of their article (what does this mean?). If published, this will include your full peer review and any attached files.

Reviewer #1: **Yes: **Lingfei Wu

Reviewer #2: No

---

## [Author Response · Author response to Decision Letter 0]

31 May 2022

please find attached the Response to Reviewers in the revised submission

---

## [Editor Report · Decision Letter 1]

6 Jun 2022

Quantifying the rise and fall of scientific fields

PONE-D-21-36265R1

Dear Dr. Santolini,

We’re pleased to inform you that your manuscript has been judged scientifically suitable for publication and will be formally accepted for publication once it meets all outstanding technical requirements.

Kind regards,

Roberta Sinatra

Academic Editor

PLOS ONE

Additional Editor Comments (optional):

I would like to thank the authors for the thorough review of the manuscript - they have addressed all the reviewers' points, as well as my comments.

---

## [Editor Report · Acceptance letter]

13 Jun 2022

PONE-D-21-36265R1 

Quantifying the rise and fall of scientific fields 

Dear Dr. Santolini:

I'm pleased to inform you that your manuscript has been deemed suitable for publication in PLOS ONE. Congratulations! Your manuscript is now with our production department. 

Kind regards, 

on behalf of

Prof. Roberta Sinatra 

Academic Editor

PLOS ONE